# Polycyclic Aromatic Hydrocarbons and Pancreatic Cancer: An Analysis of the Blood Biomarker, *r*-1,*t*-2,3,*c*-4-Tetrahydroxy-1,2,3,4-tetrahydrophenanthrene and Selected Metabolism Gene SNPs

**DOI:** 10.3390/nu16050688

**Published:** 2024-02-28

**Authors:** Sierra Nguyen, Heather Carlson, Andrea Yoder, William R. Bamlet, Ann L. Oberg, Gloria M. Petersen, Steven G. Carmella, Stephen S. Hecht, Rick J. Jansen

**Affiliations:** 1Department of Public Health, North Dakota State University, Fargo, ND 58105, USA; sierra.nguyen@coyotes.usd.edu; 2Fairbanks School of Public Health, Indiana University-Purdue University Indianapolis, Indianapolis, IN 46202, USA; 3Masonic Cancer Center, University of Minnesota, Minneapolis, MN 55455, USAhecht002@umn.edu (S.S.H.); 4Department of Quantitative Health Sciences, Mayo Clinic, Rochester, MN 55905, USA

**Keywords:** PAH, PheT, urinary biomarker, dietary intake, pancreatic cancer, SNPs, metabolism

## Abstract

Exposure to polycyclic aromatic hydrocarbons (PAHs), byproducts of incomplete combustion, and their effects on the development of cancer are still being evaluated. Recent studies have analyzed the relationship between PAHs and tobacco or dietary intake in the form of processed foods and smoked/well-done meats. This study aims to assess the association of a blood biomarker and metabolite of PAHs, *r*-1,*t*-2,3,*c*-4-tetrahydroxy-1,2,3,4-tetrahydrophenanthrene (PheT), dietary intake, selected metabolism SNPs, and pancreatic cancer. Demographics, food-frequency data, SNPs, treatment history, and levels of PheT in plasma were determined from 400 participants (202 cases and 198 controls) and evaluated based on pancreatic adenocarcinoma diagnosis. Demographic and dietary variables were selected based on previously published literature indicating association with pancreatic cancer. A multiple regression model combined the significant demographic and food items with SNPs. Final multivariate logistic regression significant factors (*p*-value < 0.05) associated with pancreatic cancer included: Type 2 Diabetes [OR = 6.26 (95% CI = 2.83, 14.46)], PheT [1.03 (1.02, 1.05)], very well-done red meat [0.90 (0.83, 0.96)], fruit/vegetable servings [1.35 (1.06, 1.73)], recessive (rs12203582) [4.11 (1.77, 9.91)], recessive (rs56679) [0.2 (0.06, 0.85)], overdominant (rs3784605) [3.14 (1.69, 6.01)], and overdominant (rs721430) [0.39 (0.19, 0.76)]. Of note, by design, the level of smoking did not differ between our cases and controls. This study does not provide strong evidence that PheT is a biomarker of pancreatic cancer susceptibility independent of dietary intake and select metabolism SNPs among a nonsmoking population.

## 1. Introduction

The American Cancer Society estimates that about 62,210 people will be diagnosed with pancreatic cancer and 49,830 people will die of pancreatic cancer in 2022 [1]. Pancreatic cancer accounts for approximately 3% of all cancers in the US and 7% of all cancer deaths [1]. The 5-year survival rate across all stages is 8.5–10% [2,3]. The incidence of pancreatic cancer based on gender is 5.5 per 100,000 for men and 4.0 per 100,000 for women [2,3]. Eighty-five percent of pancreatic cancer will develop in the exocrine tissue, which makes up much of the pancreas and is responsible for enzyme production for digestion [2]. Pancreatic neuroendocrine tumors make up the remaining cases of pancreatic cancer [4]. These tumors are found in hormone-producing cells and are generally diagnosed at a younger age with a better outcome [4].

Smoking is a known risk factor for developing pancreatic cancer. Studies have found that smoking can increase the risk by 2–5 times and that 25% of pancreatic cancers are caused by smoking [5,6]. A prospective study conducted in China found that men who were regular smokers had a 25% increased relative risk compared to those who never smoked [7]. Cigarette smoke contains 80 compounds that have been identified as carcinogenic [8]. These compounds reach the bloodstream through the lungs, and some may eventually enter the pancreas [6].

Numerous studies have also found a relationship between well-done meats and an increased risk of pancreatic cancer [9,10,11,12]. Furthermore, grilled or barbecued meat showed a strong association with a nonlinear increased risk of 80% [9]. A cohort study consisting of 316,763 men and 220,539 women showed high total meat intake was associated with a 26% increased risk of pancreatic cancer for men and women [13]. In the same study, when stratified by gender, men with the highest intake of meat that was grilled/barbecued, oven broiled, or well/very well-done had significant increases in risk for pancreatic cancer of 48%, 47%, and 37%, respectively, when compared to men with the lowest intake [13]. Another large cohort study found significant associations for both men and women regarding total meat intake, red meat intake, and high cooking temperatures (*p*-trend 0.03, *p*-trend 0.02, and *p*-trend 0.02). When the study separated participants based on gender, they found that males consumed more red meat than white meat, and females consumed more white meat than red meat [7].

Polycyclic aromatic hydrocarbons (PAHs) are chemicals that occur naturally in coal, crude oil, and gasoline as byproducts of incomplete combustion. These chemicals can also be produced from tobacco burning and from high-temperature cooking of meat and other foods [10]. Exposure to PAHs occurs when one inhales cigarette smoke or consumes grilled meats or foods in which PAHs are found [10]. PAHs form when fat and juices from the grilled meat drip onto the hot surface or fire, causing flames and smoke that contain PAHs that adhere to the surface of the meat being grilled above the flame [13].

The effects of PAHs depend upon the body metabolically activating them [12]. This can occur through the metabolic formation of electrophiles including bay-region diol epoxides such as benzo[*a*]pyrene-7,8-diol-9,10-epoxide (BPDE) which then binds to DNA causing miscoding and permanent mutations [14]. Detoxification pathways compete with metabolic activation, which can result in excretion of metabolites through urine or feces [14]. Several urinary biomarkers have been identified to quantify exposure to PAHs. Metabolites of BaP are one type of carcinogen biomarker; however, there are instances when this biomarker cannot be used due to its low levels [14]. Therefore, phenanthrene has been used as an alternative PAH biomarker because it is found in higher concentrations in the environment and broiled food and its metabolite abundance in urine is therefore greater. The diol epoxide pathway metabolite of phenanthrene is hydrolyzed to *r*-1, *t*-2,3, *c*-4-tetrahydroxy-1,2,3,4-tetrahydrophenanthrene (PheT) which can be quantified in human urine [14]. In a study of smokers versus nonsmokers, PheT was shown to be higher in those who smoke (*p*-value = 0.0073), supporting its use as a potential biomarker [15]. The same study measured the concentrations of the diol metabolites Phe-3,4-D and Phe-1,2-D to assess the activation of the diol epoxide pathways [15]. They found the pathway was activated more in smokers than non-smokers (*p*-value < 0.01) [15]. Thus, PheT could make an effective biomarker for assessing the relationship between PAH exposure and metabolism and the risk of pancreatic cancer.

This study aims to investigate how PheT, used as a blood biomarker of PAH exposure and metabolic activation, is related to pancreatic cancer. By analyzing levels of PheT in the blood of patients, we can further examine if there is a link between PheT and specific dietary items that may cause cancer. We also conducted an exploratory analysis of select metabolism single nucleotide polymorphisms (SNPs) to examine genetic markers associated with pancreatic cancer and PheT measurement to identify any potential interactions.

## 2. Materials and Methods

### 2.1. Study Design

This is a clinic-based case-control study of subjects with and without pancreatic adenocarcinoma. The collection of participant data used in this study was approved by The Mayo Clinic IRB (IRB 10-001561 approved 17 March 2010; IRB 07-000794 approved 13 February 2007; IRB 08-001393 approved 8 April 2008). Informed consent was completed at the time of enrollment and prior to collecting any information. Participants were approached from May 2004 to December 2009 and consented to participate in a prospective registry at the time of their visit using a rapid ascertainment method [16]. The study approached 2473 patients with pancreatic adenocarcinoma, of whom 1691 consented to participate at the time of their visit. During the same period, 2708 potential controls were approached during a routine care visit and 1648 consented to participate. A total of 400 participants (202 cases and 198 controls) were selected for this study. Eligible cases and controls were selected based on the availability of the food frequency data and previously measured SNP data. Cases of pancreatic adenocarcinoma were confirmed through medical records, histology, or death certificates. Histology was confirmed via the internal EHR (electronic health records) system. Controls were matched to cases on age at the time of recruitment (in 5-year increments), race, sex, and region of residence (Olmsted County; three-state (MN, WI, IA); or outside of area). Those with a prior diagnosis of cancer except non-melanoma skin cancer were excluded.

### 2.2. Study Measurements

Both cases and controls provided a blood sample and information on demographic characteristics and potential risk factors. Information for all participants was obtained via electronic medical record or questionnaire from each participant which included information on sex (male or female), age (years), former smoking status (ever or never), number of packs per day, number of years of smoking, and history of diabetes (yes or no).

Blood samples of 1.5 mL per patient were collected, processed, and frozen/stored. Following accrual, samples were shipped to the University of Minnesota where analysis of PheT in plasma was performed. The measurement of PheT was in plasma from blood (fmol/mL). The [^13^C_6_]PheT (internal standard, 100 fmol) was added to a 0.5 mL aliquot of plasma, and the PheT fraction was obtained and assayed by GC-MS as described previously [17]. Briefly, the PheT-containing fraction of the plasma was obtained by a combination of solid phase extraction and HPLC collection. The fraction was dried, silvlated, and injected on GC-NICI-MS for measurement.

SNP data were generated in lymphocyte DNA. The SNP data were obtained from a genome-wide study containing 12 prospective cohorts and 8 case-control studies [18]. A total of 1293 SNPs across 154 KEGG metabolism pathway genes were selected from the 551,766 SNPs available from the previous study [18].

Participants were asked to complete a 144-item food frequency questionnaire (FFQ) that included average consumption and frequency of intake to address possible dietary associations with pancreatic cancer. The questionnaire was modeled from the New England Bladder Cancer food frequency questionnaire that was developed by the National Cancer Institute [19]. All participants were asked to think about their usual dietary intake during the 5 years prior to entering the study. To avoid biases, cases were rapidly enrolled, and questionnaires were completed at the time of confirmed diagnosis to limit recall bias of past events. Individuals who were included in this study did not report a change in diet within the previous 5 years and did not have 17 or more items missing on their questionnaires.

### 2.3. Statistical Analysis

R statistical program [20] (Version 4.1.2) was used for data analysis and visualization. A univariate analysis was performed to compare the demographic factors between cases and controls. An independent *t*-test was performed to compare continuous variables and a chi-squared test was used for categorical comparison between cases and controls. *p*-values less than 0.05 were determined to be statistically significant. All SNPs in this study were evaluated to determine significant SNP associations with case status using 5 different models (codominant, dominant, recessive, overdominant, and additive). We used a Bonferroni method to correct a *p*-value of 0.05 for multiple testing. On this list of significant SNPs, we performed an exploratory analysis to investigate potential SNP–SNP interactions. Multiple logistic regression analysis was performed on the list of combined univariate-identified significant demographic and SNP variables. We used backward stepwise selection to exclude non-significant variables in a multi-variable setting using a *p*-value cutoff of 0.05.

## 3. Results

The study consisted of 202 cases and 198 controls sampled from a clinical setting. Of the demographic variables, there was a significantly higher average BMI among cases compared to controls (28.4 vs. 27.2) and a higher percentage diagnosed with Type 2 Diabetes among cases (28.7% vs. 9.09%) (Table 1). In addition, cases were significantly more likely to have a higher PheT (fmol/mL) measurement compared to controls (p._overall_ < 0.018). Of the food components evaluated, there were significantly higher percentages of cases compared to controls in the higher consumption categories for servings of fruit. (Table 2). There was a significantly lower percentage of cases compared to controls who consumed very well-done red meats and total nitrites (Table 2). The rest of the food components remained nonsignificant. A complete table of the results is available in the Appendix A.

After correction for multiple testing, 28 SNPs were significantly associated with pancreatic cancer in at least five model types (Figure 1). The codominant, recessive, and overdominant models identified at least three SNPs each based on the Bonferroni corrected *p*-value cutoff. The most significant SNPs across all models were rs12203582 (gene: IL17F), rs566979 (gene: CAT), rs3784605 (gene: IGF1R), rs721430 (gene: IL17F), rs3020434 (gene: ESR1), rs12029406 (gene: none, chromosome: 1), rs7775047 (gene: ESR1), rs12673242 (gene: GCK), rs2064331 (gene: none, chromosome: 6), rs5275 (gene: PTGS2), rs28540420 (gene: PTGER4), rs3790844 (gene: NR5A2), and rs2017500 (gene: IGF1R).

The 28 SNPs identified as significant across more than one of the five models in Figure 1 were further investigated for potential SNP–SNP interactions (Figure 2). Given the significant number of additional tests this pairwise interaction analysis involves across the five different SNP model types, we performed this part of the analysis for exploratory purposes only. The most significant (based on Bonferroni correction) pairs of SNPs associated with pancreatic cancer will be highlighted here. An interesting pattern was observed when focusing on SNPs that most significantly interacted with more than one other SNP in this subset. A collection of five SNPs were observed to form several of the most significant interaction pairs across different SNP models. These were rs566979 (gene: CAT), rs28540420 (gene: PTGER4), rs12029406 (gene: none), rs730368 (gene: PTGER4), and rs3181077 (gene: CCR1).

A multiple logistic regression model was built which included the demographic and dietary components previously identified from the literature and the most significant SNPs identified in Figure 1 (Table 3). Each one of the demographic and dietary variables remained significant or marginally significant. The following SNPs also remained significant in this multiple regression model: rs12203582 (gene: IL17F), rs566979 (gene: CAT), rs3784605 (gene: IGF1R), and rs721430 (gene: IL17F). When demographic/dietary variable—SNP interaction terms were evaluated, most were nonsignificant. However, the interaction between “very well-done meat” and “rs721430” was evaluated, the interaction term was significant, but the single “rs721430” variable was no longer significant so the interaction was removed from the presented table.

## 4. Discussion

The purpose of this study was to investigate how PheT, used as a blood biomarker, represents a measure of dietary exposure to PAH (polycyclic aromatic hydrocarbons) and their association with pancreatic cancer. After running statistical analyses, we found an association between PheT and case status, but not after adjustment. Therefore, we did not observe evidence that PheT could be used as an independent biomarker.

After the multiple regression analysis, our results indicated a significant difference in the consumption of very well-done red meat between the cases and controls (*p*-value = 0.0019). Previous studies have also found a correlation between intake of very well-done red meats and pancreatic cancer. Stolzenberg-Solomon et al. found that a high intake of very well-done red meat increased developmental risk by 37% [13]. A study by Anderson et al. found that consumption of other types of well-done meats including pork, bacon, grilled chicken, and pan-fried chicken resulted in an increased risk of pancreatic cancer [10]. A cohort study found that those who consumed high amounts of barbequed meat had a 24% increased risk of developing pancreatic cancer [7]. The same study found a significant association between red meat and pancreatic cancer (*p*-trend 0.02) [7]. The correlation between meats/preparation style and pancreatic cancer in these studies helps support our findings. Given that cases are enrolled in the study at or near the time of diagnosis of pancreatic cancer, it is possible that dietary intake is altered because of disease processes or symptoms [21]. Patients may alter what they are eating to avoid discomfort and this could be reflected in the higher intake of fruits and vegetables and lower intake of very well-done red meat.

There is a known association between pancreatic cancer and diabetes, with diabetes being identified as the third leading risk factor behind smoking and obesity [22,23]. A significant variation was observed between those patients with diabetes and PheT concentration in plasma (Table 2. OR 3.84). The relationship between diabetes and pancreatic cancer is supported by previous studies that have found that patients with diabetes had a 1.5–2.0 times increased risk of developing pancreatic cancer [22,23]. The relationship between the two diseases may explain the effects of dietary exposure resulting in an increased concentration of PheT in plasma.

Our study also identified a very small but significant association between total nitrites and a decreased risk of pancreatic cancer (OR = 0.00). Nitrites are a precursor to *N*-nitroso compounds (NOC) which are carcinogenic in multiple animal models [24]. When metabolically activated they create DNA adducts and single-strand breaks [11]. A case-control study completed in Iowa found that a high intake of nitrites derived from animals was positively associated with pancreatic cancer (*p*-trend = 0.02 for men, *p*-trend = 0.01 for women) [9]. The same study found no significant association with plant nitrites for both men and women. A prospective cohort studies did not observe a significant association between total nitrite and nitrate intake and pancreatic cancer [11,25]. There have been positive associations reported between nitrites and other types of cancers as well. A meta-analysis comparing nitrites with numerous cancers found significant associations for glioma and thyroid cancer but no significant association with other cancers [26,27]. The inconsistent findings across previous studies suggest that the measurement of the nitrites source will be important to consider in future studies.

Several of the investigated SNPs were observed to have a significant association with pancreatic cancer. Rs12203582 and rs721430 are in the Interleukin 17F (IL17F) gene which is an effector cytokine of innate and adaptive immune system involved in antimicrobial host defense and maintenance of tissue integrity [28]. Rs566979 is in the Catalase (CAT) gene that serves to protect cells from the toxic effects of hydrogen peroxide. It promotes the growth of cells including T-cells, B-cells, myeloid leukemia cells, melanoma cells, mastocytoma cells, and normal and transformed fibroblast cells [29]. Rs3784605 is in the Insulin-Like Growth Factor 1 Receptor (IGF1R) gene which is a receptor tyrosine kinase that mediates actions of insulin-like growth factor 1 (IGF1). It binds IGF1 with high affinity and IGF2 and insulin (INS) with a lower affinity. The activated IGF1R is involved in cell growth and survival control. IGF1R is crucial for tumor transformation and survival of malignant cells [30]. 

BMI was similar among cases (mean of 28.4) and controls (mean of 27.2) but showed a significant difference (OR 1.04) but not in adjusted models. A previous study found a positive correlation between BMI and the intake of red meats [31]. BMI is an independent risk factor for pancreatic cancer [32,33]. From a public health standpoint, BMI is interesting as it is modifiable and can change with lifestyle changes including exercise and diet, thereby reducing the risk of disease.

Although there was a significant difference regarding very well-done meat, no significant results were observed for the rest of the meat categories. Previous literature has suggested a positive correlation between the total amount of all meat consumed and pancreatic cancer (*n* = 537,203, *p* trend = 0.004, 95% CI = 1.02–1.56) [31]. Other studies’ findings have supported the relationship of increased risk between red meat consumption and pancreatic cancer (*p* trend = <0.001, *p*-value = 0.039) [7,13]. Another study found a significant difference between cases and controls on the consumption of chicken and pan-fried chicken (*p*-values < 0.001 and <0.03, respectively) [11]. Further research looking into other meat categories would be beneficial for understanding the importance.

Due to the design of our study, we did not observe significant differences regarding the number of cigarette packs smoked between the groups. Along with smoking, our results found no significant difference among the groups regarding intake of dietary mutagens: MeIQx, PhIP, DiMeIQx, and BaP. Previous studies have found an increased risk of pancreatic cancer with some of these dietary mutagens suggesting there may be a possible relationship [10,11,13,16]. The inconsistent results between our study and others suggest that more research should be completed to understand these potential associations.

The controlled way the specimens were collected and tested was important to the study to avoid introducing contamination that may skew the results. Control matching was also important to the study to accurately compare the two groups. Finally, the addition of a multiple regression analysis helped to further eliminate remaining insignificant variables.

The size of our sample was relatively small with regard to the number of variables that were tested. Although we had a desired 0.80 power amongst the groups, a larger sample size may have produced more accurate results with better insight into the associations between dietary exposure and the presence of PheT in the specimens. Another limitation of the study included recall bias specific to recall of dietary patterns when answering the food frequency questionnaire. Factors that may have contributed to this potential bias are a memory of consumption patterns as well as underreporting of certain dietary consumption due to stigmas associated with certain foods. For example, those participants with diabetes may have responded based on their dietary recommendations rather than their actual diets. In attempts to reduce this bias, cases were enrolled and completed the food frequency questionnaire at the time of diagnosis or close to the time of diagnosis. Additional limitations surrounding the food frequency questionnaire included participants who did not answer areas of the questionnaire, which could have biased results among cases and controls. For example, there were approximately 128 missing values for the FFQ for meat consumption and preparation method (92.2% being cases and 7.8% being controls). Future studies estimating dietary exposure would benefit considerably by ensuring the FFQ is completed thoroughly by the participant. Residual confounding must also be considered as smoking is a strong risk factor for pancreatic cancer, although the analyses did not show statistical significance. A previous study showed a positive association between the presence of PheT and cigarete smoking, observing a higher frequency of PheT in smokers versus lifelong never smokers [34].

## 5. Conclusions

Overall, the study showed a small association between those cases with pancreatic adenocarcinoma and the levels of PheT found in their plasma, suggesting a level of exposure to PAHs and a potential role in pancreatic adenocarcinoma. Our study analysis identified significant differences amongst factors including type 2 diabetes, very well-done red meats, PheT value, and several metabolism SNPs. A significant association between PheT and pancreatic cancer was not observed after adjusting for other factors in our multiple regression model. Future studies should further analyze the relationship between dietary intake and PheT levels using a larger sample size and conduct laboratory experiments to assess the biological and metabolic mechanisms involved in this relationship.

## Figures and Tables

**Figure 1 nutrients-16-00688-f001:**
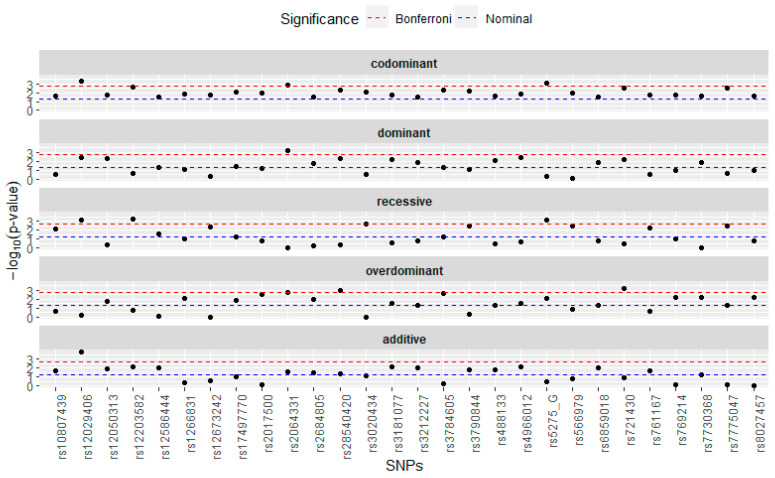
SNPs associated with pancreatic cancer adjusted for PheT. Each point in the plot represents the −log10 (*p*-value) for each SNP for each of five models. The dashed blue line indicates using a *p*-value cutoff equal to 0.05 and dashed red line indicates using a Bonferroni corrected *p*-value cutoff.

**Figure 2 nutrients-16-00688-f002:**
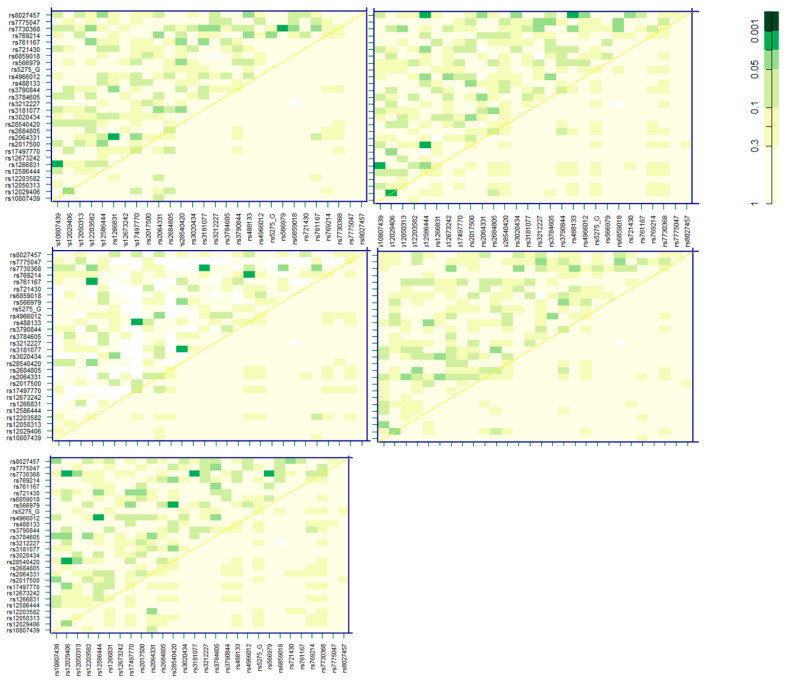
Pancreatic cancer-associated SNP interaction heatmaps displaying log-likelihood ratio test *p*-values. Models adjusted for PheT. Upper triangle presents interaction *p*-values, diagonal presents crude SNP *p*-values, and lower triangle presents LRT comparing best two-SNP additive likelihood to the best of the single-SNP models. Order of models left to right and top to bottom: codominant, dominant, recessive, overdominant, and additive.

**Table 1 nutrients-16-00688-t001:** Select demographic variables and mean PheT by case status.

	Control	Case	OR [95% CI]	p._overall_
	N = 198 Mean (SD) or N(%)	N = 202 Mean (SD) or N(%)		
PheT (fmol/mL)	16.9 (11.1)	21.5 (25.1)	1.01 [1.00;1.03]	0.018
Smoked:				0.847
Never	101 (51.0%)	106 (52.5%)	Ref.	
Former	97 (49.0%)	96 (47.5%)	0.94 [0.64;1.40]	
Sex:				0.815
Female	82 (41.4%)	87 (43.1%)	Ref.	
Male	116 (58.6%)	115 (56.9%)	0.93 [0.63;1.39]	
Age	68.0 (10.7)	68.5 (10.4)	1.01 [0.99;1.02]	0.600
BMI	27.2 (4.67)	28.4 (5.66)	1.04 [1.00;1.09]	0.029
Pack_Years	10.6 (17.4)	9.65 (17.1)	1.00 [0.99;1.01]	0.581
Type_2_Diabetes:				<0.001
No	180 (90.9%)	144 (71.3%)	Ref.	
Yes	18 (9.09%)	58 (28.7%)	4.03 [2.27;7.14]	

Abbreviations: N = number, % = percent, and SD = standard deviation, p.overall = *p*-value for the trend test comparsion, Ref. = Reference group for all comparisons.

**Table 2 nutrients-16-00688-t002:** Univariate analysis among food component variables by case status. Food components are adjusted for calorie intake and are presented as per 1000 calories.

	Control	Case	OR [95% CI]	p._overall_
	N = 188 Mean (SD)	N = 93 Mean (SD)		
PheT (fmol/mL)	10.3 (7.65)	15.2 (19.9)	1.03 [1.01;1.05]	0.023
Very_Well_Done_Red_Meat(g)	6.80 (7.09)	4.10 (4.09)	0.91 [0.87;0.97]	<0.001
Total_Nitrite	0.08 (0.09)	0.05 (0.06)	0.00 [0.00;0.15]	0.001
Fruits/Vegetables(servings)	2.07 (1.14)	2.46 (1.43)	1.28 [1.05;1.55]	0.022

Abbreviations: N = number, SD = standard deviation, p.overall = *p*-value for the trend test comparison.

**Table 3 nutrients-16-00688-t003:** Multivariable logistic regression model for pancreatic cancer including significant demographic, food/kilocalories, and SNP variables.

	OR	Lower 95% CI	Upper 95% CI	*p*-Value
Type_2_Diabetes	6.26	2.83	14.46	9 × 10^−6^
BMI	1.04	0.97	1.11	0.025
PheT	1.01	0.99	1.03	0.232
Very_Well_Done_Red_Meat(g)	0.9	0.83	0.96	0.0019
Fruits/Vegetables (servings)	1.35	1.06	1.73	0.0155
Total Nitrites	0.01	0	0.81	0.0606
Recessive (rs12203582)	4.11	1.77	9.91	0.0012
Recessive (rs566979)	0.26	0.06	0.85	0.0421
Overdominant (rs3784605)	3.14	1.69	6.01	0.0004
Overdominant (rs721430)	0.39	0.19	0.76	0.0068

## Data Availability

The de-identified datasets generated and analyzed for this study can be requested by contacting the corresponding author. The data are not publicly available due to privacy.

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
