# Peer review of "Polycyclic Aromatic Hydrocarbons and Pancreatic Cancer: An Analysis of the Blood Biomarker, r-1,t-2,3,c-4-Tetrahydroxy-1,2,3,4-tetrahydrophenanthrene and Selected Metabolism Gene SNPs"

_nutrients, 2024, doi:10.3390/nu16050688_

Round 1
Reviewer 1 Report
Comments and Suggestions for Authors
The manuscript investigating the potential association between biomarkers of PAH and pancreatic cancer is addressing an interesting research questions. However, the manuscript suffer of many unclarity including a confusing research question and lack of information in the method section making it of low scientific soundness.
Please see specfic comments:
Method section:
The method section must be restructured to give a better flow and clarity of the study design, measuraments of exposure, covariates and confounders. Morevoer, a flow chart of the study participants will help the reader to visualize better the design and the inclusion. I highly reccomend to expand the section and use subheadings (study design, measuraments of exposure and so on). The statistical methods needs also a clarification of which method was used for.
Please see specific comments
line 107: how the information about the participants were retrieved. Was the same for cases and controls?
lines 109-112: The information about the measurements of PhT in plasma are described very shortly. Since this is the exposure I would reccomend to add more information on the analytic approach e.g. is this a validated method?
lines 113-114: please explain what it means "selective sampling"
lines 119-121: the author go back to the measuraments.
lines 139-145: Please expand the section and use a subheading
Comments on the Quality of English LanguageEnglish is ok. However, I think that the manuscript needs to be re-written to clarify important points which are missing in this version.
Author Response
Reviewer 1
The manuscript investigating the potential association between biomarkers of PAH and pancreatic cancer is addressing an interesting research questions. However, the manuscript suffer of many unclarity including a confusing research question and lack of information in the method section making it of low scientific soundness.
Response: Thank you for your helpful suggestions.
Please see specfic comments:
Method section:
The method section must be restructured to give a better flow and clarity of the study design, measuraments of exposure, covariates and confounders. Morevoer, a flow chart of the study participants will help the reader to visualize better the design and the inclusion. I highly reccomend to expand the section and use subheadings (study design, measuraments of exposure and so on). The statistical methods needs also a clarification of which method was used for.
Response: Thank you for the suggestions. Subheadings added and methods section text rearranged.
- Materials and Methods
Study design
This is a clinic-based case-control study of subjects with and without pancreatic adenocarcinoma. This study was approved by The Mayo Clinic IRB in May 2004. Informed consent was completed at the time of enrollment and prior to collecting any information. Participants were approached from May 2004 to December 2009 and consented to participate in a prospective registry at the time of their visit using a rapid ascertainment method[16,17]. The study approached 2,473 patients with pancreatic adenocarcinoma, of whom 1,691 consented to participate at the time of their visit. During the same period, 2,708 potential controls were approached during a routine care visit and 1,648 consented to participate. A total of 400 participants (202 cases and 198 controls) were selected for this study. Eligible cases were selected based on availability of the food frequency data and previously measured SNP data. Controls were matched to cases on age at time of recruitment (in 5-year increments), race, sex, and region of residence (Olmsted County; three-state (MN, WI, IA); or outside of area). Those with prior diagnosis of cancer except non-melanoma skin cancer were excluded.
Study measurements
Information for all participants was obtained via electronic medical record and questionnaire from each participant included: sex (male or female), age (years), former smoking status (ever or never), number of packs per day, number of years of smoking, history of diabetes (yes or no), and measurement of PheT in plasma from blood (fmol/ml). [13C6]PheT (internal standard, 100 fmol) was added to a 0.5 ml aliquot of plasma and the PheT fraction was obtained[18] and assayed by GC-MS/MS as described previously[19]. Briefly, the PheT-containing fraction of the plasma was eluted with 5mL 40% methonal in H20 and 5mL 90:5:5 H20/methanol/NH4OH. The fraction was transferred to a nylon filter HPLC vial with 30 micrograms of 2,7-dihydroxynaphtalene and the eluant dried with occasional mixing and 4 microliters injected on CG-NICI-MS for measurement.
Both cases and controls provided a blood sample and information on demographic characteristics and potential risk factors. Blood samples of 1.5 ml per patient were collected, processed, and frozen/stored. Following accrual, samples were shipped to the University of Minnesota where analysis of PheT in plasma was performed. Cases of pancreatic adenocarcinoma were confirmed through medical records, histology, or death certificate. Histology was confirmed via the internal EHR (electronic health records) system and SNP data were generated on lymphocyte DNA. The SNP data was obtained from a genome wide study containing 12 prospective cohorts and 8 case-control studies[20] 1,293 SNPs across 154 KEGG metabolism pathway genes were selected from the 551,766 SNPs available from the previous study [20]..
Participants were asked to complete a 144-item food frequency questionnaire that included average consumption and frequency of intake to address possible dietary associations with pancreatic cancer. The questionnaire was modeled from the New England Bladder Cancer food frequency questionnaire that was developed by the National Cancer Institute[21]. All participants were asked to think about their usual dietary intake during the 5 years prior to entering the study. To avoid biases, cases were rapidly enrolled, and questionnaires were completed at the time of confirmed diagnosis to limit recall bias of past events. Individuals who were included in this study did not report a change in diet within the previous 5 years and did not have 17 or more items missing on their questionnaires.
Statistical Analysis
R statistical program (Version 4.1.2) was used for data analysis and visualization. A univariate analysis was performed to compare the demographic factors between cases and controls. An independent t-test was performed to compare continuous variables and a chi-squared test was used for categorical comparison between cases and controls. P-values less than 0.05 were determined to be statistically significant. All SNPs in this study were evaluated to determine significant SNP associations with case status using 5 different models (codominant, dominant, recessive, and overdominant, additive). We used a Bonferroni method to correct a p-value of 0.05 for multiple testing. On this list of significant SNPs we performed an exploratory analysis to investigate potential SNP-SNP interactions. Multiple logistic regression analysis was performed on the list of combined univariate-identified significant demographic and SNPs variables. We usedbackward stepwise selection to exclude non-significant variables in a multi-variable setting using a p-value cutoff of 0.05.
Please see specific comments
line 107: how the information about the participants were retrieved. Was the same for cases and controls?
Response: Modified sentences as follows: “Information for all participants was obtained via electronic medical record and questionnaire…” to clarify.
lines 109-112: The information about the measurements of PhT in plasma are described very shortly. Since this is the exposure I would reccomend to add more information on the analytic approach e.g. is this a validated method?
Response: Two published references for the process have already been included. We additionally added a brief explanation so it reads as follows:
“[13C6]PheT (internal standard, 100 fmol) was added to a 0.5 ml aliquot of plasma and the PheT fraction was obtained[18] and assayed by GC-MS/MS as described previously[19]. Briefly, the PheT-containing fraction of the plasma was eluted with 5mL 40% methonal in H20 and 5mL 90:5:5 H20/methanol/NH4OH. The fraction was transferred to a nylon filter HPLC vial with 30 micrograms of 2,7-dihydroxynaphtalene and the eluant dried with occasional mixing and 4 microliters injected on CG-NICI-MS for measurement.”
lines 113-114: please explain what it means "selective sampling"
Response: We just meant that cases were selected based on data availability. We have rewritten the sentence as follows to clarify: “Eligible cases were selected based on availability of the food frequency data and previously measured SNP data”
lines 119-121: the author go back to the measuraments.
Response: Section has been rearranges to the all text on measurements is together.
lines 139-145: Please expand the section and use a subheading
Response: We have added subheadings to the entire section and have added additional details on our statistical approaches.
Reviewer 2 Report
Comments and Suggestions for Authors
The aim of this study was to determine the role of the metabolite of PAHs- PheT as a biomarker for pancreatic cancer. In this paper, grilled meats or foods were recognized as the main sources of PheT. Overall the study is well executed, the result well described and the conclusions justified. I only have a few minor points that are not clear and should be addressed.
1. pls check editing - there are a lot of double or triple spacer in the text
2. Table 3 also requires editing
3. Is it possible to add additional information about the methodology of the SNP determination?4. Consider clarifying the information about exclusion and inclusion criteria (for example other diseases present)
5. Consider discussing why pancreatic patients consume less red meat and more fruits.
6. in the discussion section, where you describe particular SNPs, it needs to be clarified that those SNPs are related to pancreatic incidence in this study group.
Author Response
Reviewer 2
The aim of this study was to determine the role of the metabolite of PAHs- PheT as a biomarker for pancreatic cancer. In this paper, grilled meats or foods were recognized as the main sources of PheT. Overall the study is well executed, the result well described and the conclusions justified. I only have a few minor points that are not clear and should be addressed.
Response: Thank you for your helpful suggestions.
- pls check editing - there are a lot of double or triple spacer in the text
Response: Corrected.
- Table 3 also requires editing
Response: Corrected.
- Is it possible to add additional information about the methodology of the SNP determination?
Response: The SNP processing is detailed in the attached referenced study (reference #20).
- Consider clarifying the information about exclusion and inclusion criteria (for example other diseases present)
Response: We have already included a statement that those with cancer exempt non-melanoma skin cancer have been excluded. With the restructuring of the methods section, we anticipate this is clearer.
- Consider discussing why pancreatic patients consume less red meat and more fruits.
Response: Added the following to the discussion:
“Given that cases are enrolled in the study at or near the time of diagnosis of pancreatic cancer, it is possible that dietary intake is altered because of disease process or symptoms. Patients may alter what they are eating to avoid discomfort and this could be reflected in the higher intake of fruits and vegetables and lower intake of very well done read meat.”
- in the discussion section, where you describe particular SNPs, it needs to be clarified that those SNPs are related to pancreatic incidence in this study group.
Response: Added.
Round 2
Reviewer 1 Report
Comments and Suggestions for Authors
The manuscript has an interesting research question. However, the description of statistical methods and results must be improved. Overall, there is lack of clarity and scientific soundness that makes this manuscript difficult to read.
Author Response
We have extensively edited the methods section and made changes to the results section to improve clarity.
Reviewer 2 Report
Comments and Suggestions for Authors
The corrected manuscript can be considered for publication.
Author Response
Reviewer 2
The aim of this study was to determine the role of the metabolite of PAHs- PheT as a biomarker for pancreatic cancer. In this paper, grilled meats or foods were recognized as the main sources of PheT. Overall the study is well executed, the result well described and the conclusions justified. I only have a few minor points that are not clear and should be addressed.
- pls check editing - there are a lot of double or triple spacer in the text
Response: Fixed. Thank you.
- Table 3 also requires editing
Response: Fixed. Thank you.
- Is it possible to add additional information about the methodology of the SNP determination?
Response: We have edited the methods section to improve clarity. Subheadings added and methods section text rearranged.
- Materials and Methods
Study design
This is a clinic-based case-control study of subjects with and without pancreatic adenocarcinoma. This study was approved by The Mayo Clinic IRB in May 2004. Informed consent was completed at the time of enrollment and prior to collecting any information. Participants were approached from May 2004 to December 2009 and consented to participate in a prospective registry at the time of their visit using a rapid ascertainment method[16,17]. The study approached 2,473 patients with pancreatic adenocarcinoma, of whom 1,691 consented to participate at the time of their visit. During the same period, 2,708 potential controls were approached during a routine care visit and 1,648 consented to participate. A total of 400 participants (202 cases and 198 controls) were selected for this study. Eligible cases were selected based on availability of the food frequency data and previously measured SNP data. Controls were matched to cases on age at time of recruitment (in 5-year increments), race, sex, and region of residence (Olmsted County; three-state (MN, WI, IA); or outside of area). Those with prior diagnosis of cancer except non-melanoma skin cancer were excluded.
Study measurements
Information for all participants was obtained via electronic medical record and questionnaire from each participant included: sex (male or female), age (years), former smoking status (ever or never), number of packs per day, number of years of smoking, history of diabetes (yes or no), and measurement of PheT in plasma from blood (fmol/ml). [13C6]PheT (internal standard, 100 fmol) was added to a 0.5 ml aliquot of plasma and the PheT fraction was obtained[18] and assayed by GC-MS/MS as described previously[19]. Briefly, the PheT-containing fraction of the plasma was eluted with 5mL 40% methonal in H20 and 5mL 90:5:5 H20/methanol/NH4OH. The fraction was transferred to a nylon filter HPLC vial with 30 micrograms of 2,7-dihydroxynaphtalene and the eluant dried with occasional mixing and 4 microliters injected on CG-NICI-MS for measurement.
Both cases and controls provided a blood sample and information on demographic characteristics and potential risk factors. Blood samples of 1.5 ml per patient were collected, processed, and frozen/stored. Following accrual, samples were shipped to the University of Minnesota where analysis of PheT in plasma was performed. Cases of pancreatic adenocarcinoma were confirmed through medical records, histology, or death certificate. Histology was confirmed via the internal EHR (electronic health records) system and SNP data were generated on lymphocyte DNA. The SNP data was obtained from a genome wide study containing 12 prospective cohorts and 8 case-control studies[20] 1,293 SNPs across 154 KEGG metabolism pathway genes were selected from the 551,766 SNPs available from the previous study [20]..
Participants were asked to complete a 144-item food frequency questionnaire that included average consumption and frequency of intake to address possible dietary associations with pancreatic cancer. The questionnaire was modeled from the New England Bladder Cancer food frequency questionnaire that was developed by the National Cancer Institute[21]. All participants were asked to think about their usual dietary intake during the 5 years prior to entering the study. To avoid biases, cases were rapidly enrolled, and questionnaires were completed at the time of confirmed diagnosis to limit recall bias of past events. Individuals who were included in this study did not report a change in diet within the previous 5 years and did not have 17 or more items missing on their questionnaires.
Statistical Analysis
R statistical program (Version 4.1.2) was used for data analysis and visualization. A univariate analysis was performed to compare the demographic factors between cases and controls. An independent t-test was performed to compare continuous variables and a chi-squared test was used for categorical comparison between cases and controls. P-values less than 0.05 were determined to be statistically significant. All SNPs in this study were evaluated to determine significant SNP associations with case status using 5 different models (codominant, dominant, recessive, and overdominant, additive). We used a Bonferroni method to correct a p-value of 0.05 for multiple testing. On this list of significant SNPs we performed an exploratory analysis to investigate potential SNP-SNP interactions. Multiple logistic regression analysis was performed on the list of combined univariate-identified significant demographic and SNPs variables. We usedbackward stepwise selection to exclude non-significant variables in a multi-variable setting using a p-value cutoff of 0.05.
- Consider clarifying the information about exclusion and inclusion criteria (for example other diseases present)
Response: Clarified as follows: Eligible cases were selected based on availability of the food frequency data and previously measured SNP data. Controls were matched to cases on age at time of recruitment (in 5-year increments), race, sex, and region of residence (Olmsted County; three-state (MN, WI, IA); or outside of area). Those with prior diagnosis of cancer except non-melanoma skin cancer were excluded.
- Consider discussing why pancreatic patients consume less red meat and more fruits.
Response: Added the following to the discussion section:
Given that cases are enrolled in the study at or near the time of diagnosis of pancreatic cancer, it is possible that dietary intake is altered because of disease process or symptoms. Patients may alter what they are eating to avoid discomfort and this could be reflected in the higher intake of fruits and vegetables and lower intake of very well done read meat.
- in the discussion section, where you describe particular SNPs, it needs to be clarified that those SNPs are related to pancreatic incidence in this study group.
Response: Added the following to the discussion section:
Several of the investigated SNPs were observed to have a significant association with pancreatic cancer.